# Gender and Work Experience as Moderators of Relations between Management Level, Physical Activity, Eating Attitudes, and Social Skills of Managers during the COVID-19 Pandemic

**DOI:** 10.3390/nu15194234

**Published:** 2023-09-30

**Authors:** Dominika Wilczyńska, Anna Hryniewicz, Magdalena Jaroch-Lidzbarska, Konrad Hryniewicz, Mariusz Lipowski

**Affiliations:** 1Faculty of Physical Education, Gdansk University of Physical Education and Sport, 80-336 Gdansk, Poland; anna.hryniewicz@awf.gda.pl (A.H.); magdalena.jaroch-lidzbarska@awf.gda.pl (M.J.-L.); 2Department of Marketing and Quantitative Methods, Faculty of Management and Quality Science, Gdynia Maritime University, 81-225 Gdynia, Poland; metodolog.pl@gmail.com; 3Faculty of Social and Humanities, WSB Merito University, 80-266 Gdansk, Poland; mlipowski@wsb.gda.pl

**Keywords:** social skills, physical activity level, gender, eating attitudes, managers, COVID-19 pandemic

## Abstract

Being employed in a managerial position is often associated with maintaining high standards in many aspects of life. Many leaders pay attention to their physical activity, eating habits, and social skills. The onset of the COVID-19 pandemic brought additional difficulties to the already-demanding job of managing people and forced managers to make many changes to their daily functioning at work. The main goal of this study was to establish whether Gender, Experience, and Management Level influenced respondents’ healthy behaviors (eating attitudes and physical activity) or soft skills during the COVID-19 pandemic. This study was carried out during the COVID-19 pandemic with a sample of 348 managers from a variety of companies (*n* = 222 women, *n* = 126 men) with different levels of experience and responsibility. The authors used the 26-item Eating Attitudes Test (EAT-26), four questions from the Physical Activity Objectives Questionnaire, and a self-authored soft skills questionnaire. The results showed that, compared to females, males were characterized by lower levels on all three EAT-26 scales: Bulimia and Food Preoccupation, Oral Control, and Dieting. On the other hand, male respondents who held high managerial positions were characterized by high levels of Dieting, Oral Control, Bulimia, and Food Preoccupation. This analysis provides insights that may help improve the quality of life of employees; however, further research is needed to investigate the direct influence of managers on employees in different industries.

## 1. Introduction

During the worldwide crisis caused by the pandemic, almost all workplaces had to make significant changes quickly—in particular; switching to remote working [1]. For a variety of reasons, managers are a professional group that may have been particularly vulnerable to the consequences of the COVID-19 pandemic. The feeling of a real threat to their health, considerable distress about their work, and the need for a sudden reorganization of the workplace for subordinates could have been sources of significant anxiety for this group, in addition to the more typical demands of the role of manager. Holding a managerial position is often associated with maintaining high standards of appearance, diet, physical activity (PA), and social skills. A manager often wishes to set a good example as part of their management of human resources. For this reason, physical fitness is considered an excellent way to build physical, psychological, and social resiliency among managers and is therefore important for maintaining a healthy lifestyle. Another significant motivation for undertaking PA is that physically fit leaders have been found to have increased stamina and mental focus [2,3,4,5]. Moreover, leaders who exercise regularly might benefit from immediate post-exercise effects, such as improvements in information processing, error recognition, executive function, and decision-making [6]. Additionally, engaging in physical activity, particularly at a moderate intensity, demonstrates a positive correlation with both health perception and mood. Furthermore, it has been shown to contribute positively to well-being by enhancing recovery experiences [7,8]. There is emerging evidence suggesting that regular physical activity might be linked to a decreased likelihood of experiencing severe outcomes from COVID-19. These findings underscore the protective role of adequate physical activity as a valuable public health strategy, offering potential advantages in reducing the risk of severe COVID-19 cases. Notably, individuals who regularly participate in physical activities exhibited lower rates of hospitalization, severe illness, and COVID-19-related fatalities in comparison to their less active counterparts. Recent studies [9,10] have reported that adults with high and moderate levels of physical activity experienced significantly better outcomes upon contracting COVID-19 than those with low levels of activity. With all the above benefits in mind, we were interested in how experience and management level could affect the PA of managers.

When examining the impact of eating behaviors on well-being during a pandemic period, certain studies have revealed that participants exhibited subpar well-being, inadequate levels of physical activity, and moderate scores in terms of healthy eating. Notably, an increase in physical activity and the adoption of healthier eating habits were linked to improved well-being, while a sedentary lifestyle correlated with a decline in well-being [11]. Consequently, our curiosity extended to eating attitudes, especially within the context of potential eating disorders among leaders. This additional focus on eating attitudes seamlessly integrates with our understanding of leaders’ well-being, underscoring the intricate connections between physical activity, dietary preferences, and mental health. According to the leading eating disorder treatment center in the USA, the Center for Discovery, research has shown that many high-level leaders experience significant stress and, as a result, use comfort food as a coping mechanism. Engaging in comfort eating puts individuals at risk of developing bulimia or binge eating disorders. Attention has also been paid to women who have high-achieving personalities and a strong desire to accomplish their goals. Women who are high achievers are more likely to hold higher positions in their fields and face significant pressure, especially from the gender gap in society, to achieve the ‘thin ideal.’ Constantly being pressured and reminded that they must fulfill physical and societal expectations could seriously harm their feelings of self-worth. Poor self-worth can result in poor self-image and low self-esteem, which can drive women to engage in unhealthy eating behaviors, resulting in eating disorders [12]. Potential differences in eating attitudes between the genders were also a focus of our attention.

Empirical studies indicate that managers spend up to 70% of their time interacting and communicating with other people. Managers are in daily contact with individual people but also with group representatives, businesses, and the public. This can be very demanding. Empathy, assertiveness, the ability to motivate and to listen, and other social skills are crucial for the effective management of human resources [13]. Since the early years of the twentieth century, the individual characteristics of leaders have been systematically researched in order to find and explain the factors that distinguish the exemplary manager and to help managers lead more effectively [14]. Rosiński states that a leader in a management situation is the person who has the greatest influence and power to motivate as well as to help employees be self-motivated by passion, feel part of a group, and perceive the leader as a model. A manager should not only be task- and management-oriented but should also focus on subordinates [15], as many theories of leadership emphasize the importance of a leadership style that balances people’s needs with production and considers the interaction between the leader’s personality and the control aspect of the situation as the most optimal approach to leadership.

Our research focused on certain soft skills of managers. Choudary and Ponnuru indicate that “soft skills” are often associated with a person’s Emotional Intelligence Quotient. Thus, soft skills are linked with factors central to relationships with other people, such as personality traits, social competence, communication, language, personal habits, interpersonal skills, managing people, and leadership [15]. Cimatii adds that the term “soft skills” is an indicator of all the competencies not directly connected to a specific task that pertains to relationships with other people in the organization. The whole quality of an enterprise depends strongly on the soft skills possessed by personnel at every level [16].

This paper sought to describe managers of different levels and experiences and their cooperation with employees during the pandemic. Furthermore, we wanted to investigate whether gender, management level, and experience were associated with a manager’s soft skills and PA levels. Therefore, we formulated the following research questions: (1) How does gender affect the eating attitudes and PA of managers? (2) Are experience and management level mediators of managers’ PA levels and soft skills?

## 2. Materials and Methods

### 2.1. Sample

This study group consisted of managers from Polish companies invited to participate in this study using the snowball sampling procedure. Prior to the start of this study, participants gave their written consent to participate. Data were collected in 2022, during the fourth and fifth waves of the pandemic. This study was conducted using an online questionnaire.

This study involved *N* = 348 managers (*n* = 222 women; *n* = 126 men). The majority of respondents (58%) were top-level managers, 25% were mid-level, and 17% were first-line managers; 175 of the respondents had many years’ experience (*M* = 5.86), and slightly fewer (173) had fewer years’ experience (*M* = 5.53). The mean age of the participants was *M* = 50.72 (*SD* = 9.66).

### 2.2. Instruments

The research survey began with a questionnaire constructed by the authors of this study, consisting of seven questions concerning managers’ soft skills, three questions regarding PA, and five questions collecting information about the attitude of managers towards COVID-19. Below are the specific elements of the researched social skills, together with sample questions.
(a)Emotional Interest (‘Do you have more interest in your subordinates’ emotional lives during the COVID-19 pandemic than before it?’)(b)Motivation (‘Do you motivate your subordinates with words, e.g., “You can do this,” “I believe in you”, more often during the COVID-19 pandemic than before it?’)(c)Supporting (‘Do you ask your subordinates questions such as “What is your biggest fear?” or “How can I help you?” more often during the COVID-19 pandemic than before it?’)(d)Health Interest (‘Has the COVID-19 pandemic increased your interest in the health of your subordinates?’)

We used four questions from the Physical Activity Objectives Questionnaire, in its Polish version by Lipowski and Zaleski [17], to assess PA over the course of a month. Participants responded to the following questions: (1) ‘Do you participate in classes (e.g., in a fitness club/gym)?’—yes/no response; (2) ’How many times a month?’—open question; (3) ‘Do you engage in physical activity on your own?’—yes/no response; (4) ‘How many times a month?’—open question.

We used the Eating Attitude Test (EAT-26), in its Polish version by Rogoza, Brytek-Matera, and Garner [18], to assess the eating behaviors of the participants. The test consists of three subscales concerning: Dieting (e.g., ‘I am preoccupied with a desire to be thinner’), Bulimia and Food Preoccupation (e.g., ‘I feel that food controls my life’), and Oral Control (e.g., ‘I cut food into small pieces’). The participants responded to 26 items on a five-point Likert scale (1—always; 5—never). Cronbach’s alpha for the Polish version of the EAT-26 was relatively good (0.85).

The online survey took place from January 2022 to December 2022. All the questionnaires used for the purpose of this online study are attached in Appendix C.

### 2.3. Statistical Analysis

All statistical analysis, tables, and figures were generated in R [19] using the kableExtra [20], ggplot [21], and jtools [22] packages. We performed a series of regression models with Gender and Experience as moderators (divided by median split *Me* = 24 years for Short and Long Experience groups). The analysis is divided into two sections: the first pertains to the Gender moderator and the second to the Management Level moderator. Furthermore, an analysis of the collected COVID-19-related variables was performed: Previous COVID-19 disease; Opinion of Diagnosis Against COVID, Being Vaccinated, and Opinion of Mandatory Vaccinations (variable characteristics are presented in Appendix A). To check the robustness of the tested models, we conducted a series of additional regression analyses, controlling for the moderating effects of the aforementioned COVID-19-related information. These analyses showed that the introduction of these variables into the models had almost no effect on the analyzed moderation effects. These results are presented in Appendix B.

## 3. Results

Below, Table 1 shows sample characteristics, descriptive statistics, and frequencies for the variables used in the analyses in gender subgroups. The table indicates that females had a lower diet score and BMI than males. BMI by category division showed that females had a rather normal weight, but males were rather overweight. There were also significant effects of education. Females often had a higher master’s degree than males, but males often had a higher PhD than females. Further, analysis of the rows of the table indicates that females had higher scores in terms of supporting, emotional, health interest, and motivating. There were no other significant differences between males and females.

### 3.1. Analysis of Gender

Regression analysis for Bulimia and Food Preoccupation showed significant results: *F*(3, 342) = 3.54, *p* < 0.05: *R*^2^ = 0.03, adj. *R*^2^ = 0.02. The analysis showed that males had lower Bulimia and Food Preoccupation than females. There was also a significant interaction between Gender and Management Level. Detailed analysis of simple interaction effects showed that there was no relation between Management Level and Bulimia and Food Preoccupation in the female group (*B* = 0.03, *t* = 0.63, *p* > 0.05, β = 0.04, 95% CI [−0.06, 0.15], *R*^2^ = 0.00). In the male group, higher Management Level was related to increased Bulimia and Food Preoccupation, *B* = 0.24, *t* = 2.95, *p* < 0.01, β = 0.26, 95% CI [0.10, 0.42], *R*^2^ = 0.07). Results are presented in Figure 1.

Further analysis of Dieting showed significant results: *F*(3, 342) = 4.31, *p* < 0.01, *R*^2^ = 0.04, adj *R*^2^ = 0.03. The analysis showed that males had lower Dieting than females. There was also a significant interaction between Gender and Management Level. Detailed analysis of simple interaction effects showed that there was no relation between Management Level and Dieting in the female group (*B* = −0.12, *t* = −1.66, *p* > 0.50, β = −0.11, 95% CI [−0.25, 0.02], *R*^2^ = 0.01. In the male group, higher Management Level was related to increased Dieting, *B* = 0.21, *t* = 2.43, *p* < 0.05, β = 0.21, 95% CI [0.04, 0.38], *R*^2^ = 0.05). Results are presented in Figure 2.

Analysis for Oral Control showed results close to significant: *F*(3, 342) = 1.92, *p* < 0.10, *R*^2^ = 0.02, adj. *R*^2^ = 0.01. The analysis showed that males had lower Oral Control than females. There was also a significant interaction between Gender and Management Level. Detailed analysis of simple interaction effects showed that there was no relation between Management Level and Oral Control in the female group (*B* = −0.02, *t* = −0.31, *p* > 0.05, β = −0.02, 95% CI [−0.12, 0.08], *R*^2^ = 0.00). In the male group, higher Management Level was related to increased Oral Control, *B* = 0.16, *t* = 2.12, *p* < 0.05, β = 0.19, 95% CI [0.04, 0.34], *R*^2^ = 0.04). Results are presented in Figure 3.

The last regression model related to the Gender moderator. The supporting social skill as a dependent variable had a significant result: *F*(3, 342) = 5.20, *p* < 0.01, *R*^2^ = 0.04, adj. *R*^2^ = 0.04. The analysis showed that males had lower Supporting than females. There was also a significant interaction between Gender and Management Level. Detailed analysis of simple interaction effects showed that there was no relation between Management Level and the Supporting social skills in the female group (*B* = 0.04, *t* = 0.25, *p* > 0.05, β = 0.02, 95% CI [−0.28, 0.31], *R*^2^ = 0.00). In the male group, higher Management Level was related to increased Supporting, *B* = 0.59, *t* = 3.04, *p* < 0.01, β = 0.26, 95% CI [−0.12, 0.65], *R*^2^ = 0.07). Results are presented in Figure 4.

There was also a model for PA level; however, this model was not significant: *F*(3, 342) = 2.10, *p* > 0.05. Results are presented in Figure 5.

### 3.2. Analysis for Experience

The regression analysis for Emotional Interest showed significant results: *F* (3, 342) = 2.69, *p* < 0.05, *R*^2^ = 0.02, adj. *R*^2^ = 0.01. The analysis showed that increased Experience and Management Level were related to a decreased level of Emotional Interest. Detailed analysis of simple interaction effects showed that there was no relation between Management Level and Emotional Interest in the Short Experience group (*B* = −0.22, *t* = −1.29, *p* > 0.05, β = −0.10, 95% CI = [−0.43, 0.23], *R*^2^ = 0.01. In the Long Experience group, higher Management Level was related to increased Emotional Interest, *B* = 0.47, *t* = 2.30, *p* < 0.05, β = 0.18, 95% CI [−0.22, 0.57], *R*^2^ = 0.03). The results are presented in Figure 6.

Further analysis for Health Interest showed significant results: *F* (3, 342) = 5.39, *p* < 0.01, *R*^2^ = 0.05, adj. *R*^2^ = 0.04. The analysis showed that increased Experience and Management Level were related to a decreased level of Health Interest. Detailed analysis of simple interaction effects showed that there was no relation between Management Level and Health Interest in the Short Experience group (*B* = −0.08, *t* = −0.58, *p* > 0.05, β = −0.04, 95% CI [−0.33, 0.24], *R*^2^ = 0.00). In the Long Experience group, higher Management Level was related to increased Health Interest, *B* = 0.59, *t* = 3.40, *p* < 0.001, β = 0.26, 95% CI [−0.09, 0.60], *R*^2^ = 0.07. Results are presented in Figure 7.

The analysis for Motivating showed significant results: *F*(3, 342) = 7.14, *p* < 0.001, *R*^2^ = 0.06, adj. *R*^2^ = 0.05. The analysis showed that increased Experience and Management Level were related to decreased Motivation. These counterintuitive results were observed due to the high collinearity of Experience (VIF = 10.30) and Management Level (VIF = 6.17). Detailed analysis of simple interaction effects showed that there was no relation between Management Level and Motivating in the Short Experience group (*B* = −0.10, *t* = −0.58, *p* > 0.05, β = −0.04, 95% CI [−0.37; 0.28], *R*^2^ = 0.00). In the Long Experience group, increased Management Level was related to increased Motivating, *B* = 0.73, *t* = 3.55, *p* < 0.001, β = 0.27, 95% CI [−0.14, 0.67], *R*^2^ = 0.07). Results are presented in Figure 8.

The last regression model related to the Management Level moderator. The Supporting social skill as a dependent variable was significant, *F*(3, 342) = 6.26, *p* < 0.001, *R*^2^ = 0.04, adj. *R*^2^ = 0.04. The analysis showed that the level of the Supporting social skill was not related to Experience; however, a higher Management Level was related to decreased Support. Nevertheless, a detailed analysis of simple interaction effects showed that there was no relation between the Management Level and the Supporting social skill in the Short Experience group. *B* = −0.11, *t* = −0.72, *p* > 0.05, β = −0.05, 95% CI [−0.37; 0.26], *R*^2^ = 0.00. In the Long Experience group, increased Management Level was related to increased Supporting, *B* = 0.67, *t* = 3.39, *p* < 0.001, β = 0.25, 95% CI [−0.14, 0.64], *R*^2^ = 0.06). Results are presented in Figure 9.

The model for PA level was significant, *F*(3, 342) = 3.28, *p* < 0.05, but only for the intercept term, *a* = 1.04, *t* = 5.33, *p* < 0.001. The predictors and interaction terms were not significant. Results are presented in Figure 10.

All regression model estimates with two main effects and interactions are presented in Table 2.

## 4. Discussion

Taking care of the mental and physical health of both one’s employees and oneself should be an important element in the life of every manager and, indeed, everybody in general. Employers have been increasingly focusing on supporting the mental health of employees, even before the pandemic [23]. Therefore, the main aim of this paper was to determine whether variables such as Gender, Experience, and Management Level influenced managers’ healthy behaviors (eating attitudes and physical activity; PA) and soft skills during the COVID-19 pandemic. This study showed some significant results. First, there were gender differences in eating attitudes and certain social skills. Males were characterized by lower levels of all three scales of EAT-26 (Bulimia and Food Preoccupation, Oral Control, and Dieting) compared to females. It is important to note that neither the women nor the men who participated in this study were found to exhibit symptoms of eating disorders. However, it is crucial to emphasize that eating attitudes in this study were evaluated using the Eating Attitude Test (EAT-26), originally designed for clinical samples to assess the propensity toward eating disorders. Therefore, exercising caution when interpreting the results is imperative [18]. Nonetheless, male respondents who held high managerial positions were characterized by high levels of Dieting, Oral Control, Bulimia, and Food Preoccupation. This particular group of respondents indicated concerns about body weight, body shape, and eating that are stereotypically expressed by women because women’s overall self-esteem is highly dependent on positive body image [24]. Though it should be noted that nowadays men attach greater importance to their appearance, which is also particularly emphasized in the corporate world, Taking care of oneself physically sets a good example for employees and is an indicator of self-control and success. On the other hand, the increasing emphasis on appearance in the corporate world, particularly among men, poses a range of challenges and concerns related to mental health, workplace culture, diversity and inclusion, productivity, and sustainability. These issues warrant careful consideration and proactive measures to ensure a balanced and equitable work environment [25,26]. Van der Put and Ellwardt [27] confirmed in their studies that healthy behaviors among both employers and colleagues can contribute to creating a culture of health in the workplace and support all employees in making healthy choices. Nickson’s [28] description of the workplace environment emphasizes that today’s society seems to be obsessed with physical attractiveness. In certain organizational contexts, the way you look can make the difference between being hired or fired. Upon contrasting the findings of our present study with those of other investigations that have delved into people’s dietary behaviors amid the pandemic, a notable deduction emerges: the COVID-19 pandemic has the potential to trigger favorable transformations in individuals’ eating habits. This supposition finds validation in data collected from a cohort of approximately 900 adults in the United States. Particularly noteworthy is the observation that individuals of younger age and higher educational attainment, with a heightened emphasis on health considerations, display a heightened likelihood of adopting positive dietary modifications [29]. Moreover, an examination of parental healthy eating behaviors uncovers a substantial gender disparity: Fathers and males exhibit significantly greater involvement in health-conscious dietary practices. This discrepancy highlights the pandemic’s overarching constructive impact on the assimilation of health-oriented dietary behaviors. This influence is encapsulated by its comprehensive effect on perceptions and behavioral patterns pertaining to dietary choices [30].

We also assumed that PA would differ between female and male respondents. However, men and women in the group of managers studied had the same levels of PA. This could be due to the current social trend of women and men having similar levels of PA and possibly dietary restrictions, as well as interventions and programs that increase PA and decrease sedentary time at work. A study by Pronk [31] points out that the workplace provides a range of opportunities for PA, using a social-ecological framework with five broad levels: personal, social, communication (including information technologies), physical, and political. The author suggests the workplace is a communal setting where these five broad levels intersect and where effective strategies and tactics work with women and men alike. On the contrary, the majority of studies indicate that there were differences between women and men in terms of their level of physical activity during COVID-19. It was found that males were more active than females and had distinct motivations for engaging in physical activity [32,33,34].

With regard to Experience and Management Level, there were a number of important observations in the current analysis. Managers with more experience and in higher positions had the strongest social skills, such as emotional and health concerns, as well as supportive and motivational attitudes toward their employees. This could be explained by their having developed a significant body of knowledge and strategies to help employees over the course of their careers be effective in difficult or crisis situations. Begtrup et al. [35], analyzing the impact of a manager-oriented intervention on the well-being of hospital and daycare workers, suggested that training managers in implementing an explicit and positive supportive approach would result in a better work environment and employee well-being. Herr et al. [36] found that ambivalent supervisor-employee relationships had an overall impact on depression, anxiety, vitality, and exhaustion among workers. At the individual and group levels, there was a consistent relationship between ambivalent leadership and higher levels of psychological distress. However, our assumptions were not confirmed: we found no effects of Experience and Management Level on PA levels. The participants in this study seemed very similar in this regard. Companies are putting more and more effort and focus into PA and exercise interventions in the workplace in order to improve work outcomes. The systematic review of White et al. [37] concluded that short and simple exercise and fitness programs, in particular, have an impact on absenteeism from work, work productivity, and financial outcomes.

This study has some limitations. There is no data concerning the character of the corporations the respondents were working for or their place of residence, which could have provided meaningful context and thereby revealed other significant correlations. This study may also have been limited by the fact that the data on PA were not very detailed; future research should look more closely at the exercise habits of the participants. Further research is needed to increase our knowledge of the relationships between these factors among managers in different industries. In particular, future research should focus on younger managers, who are understudied in the context of PA, social skills, and eating attitudes. For the quality of life of employees, this research has many benefits: improving the effectiveness and efficiency of managers can, in turn, improve employees’ professional and domestic lives.

## 5. Conclusions

This study aimed to investigate the influence of Gender, Experience, and Management Level on managers’ healthy behaviors and soft skills in the challenging context of the COVID-19 pandemic. The findings revealed noteworthy insights. Gender differences were evident, with males exhibiting lower scores across various scales of eating attitudes, though without indicating any presence of eating disorders. Interestingly, male managers in high-ranking positions displayed higher levels of specific eating attitudes, raising concerns deserving of further exploration. Surprisingly, no significant gender disparities were detected in physical activity levels among the managers studied. Examining Experience and Management Level revealed significant patterns, as more experienced and higher-ranking managers demonstrated stronger social skills, including emotional support, health concerns, and motivational attitudes towards their employees. These insights can guide strategies to promote healthy behaviors and enhance soft skills among managers, fostering resilient and effective leadership in challenging times. However, further research is encouraged to deepen our understanding of these dynamics and their implications for workplace well-being.

## Figures and Tables

**Figure 1 nutrients-15-04234-f001:**
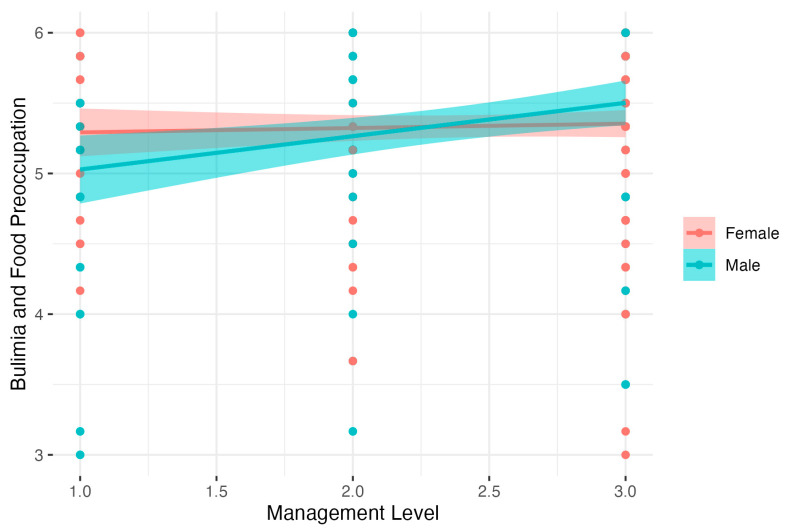
Gender as moderator of the relationship between Management Level and Bulimia and Food Preoccupation.

**Figure 2 nutrients-15-04234-f002:**
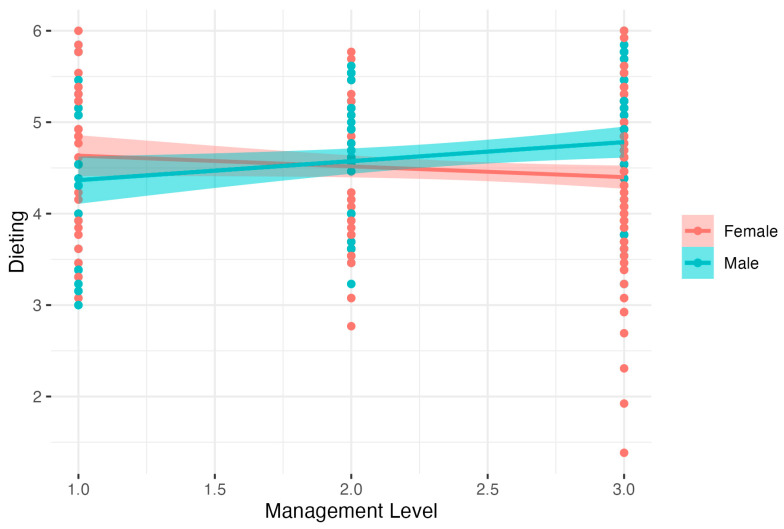
Gender as a moderator of the relationship between Management Level and Dieting.

**Figure 3 nutrients-15-04234-f003:**
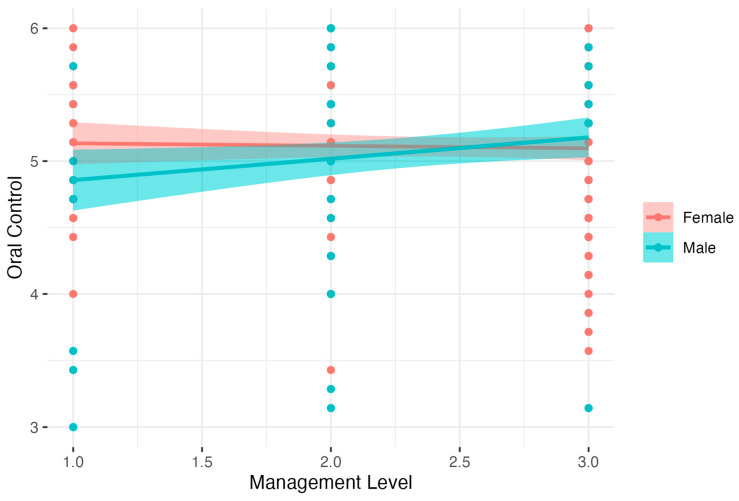
Gender as moderator of the relationship between Management Level and Oral Control.

**Figure 4 nutrients-15-04234-f004:**
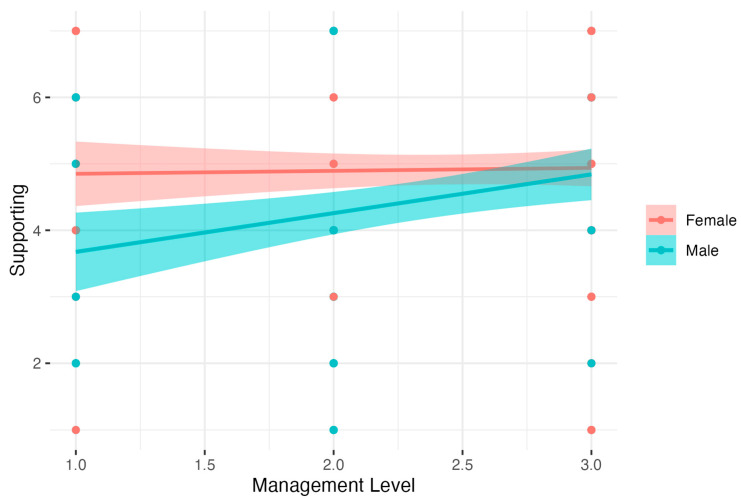
Gender as moderator of the relationship between the Management Level and the Supporting social skills.

**Figure 5 nutrients-15-04234-f005:**
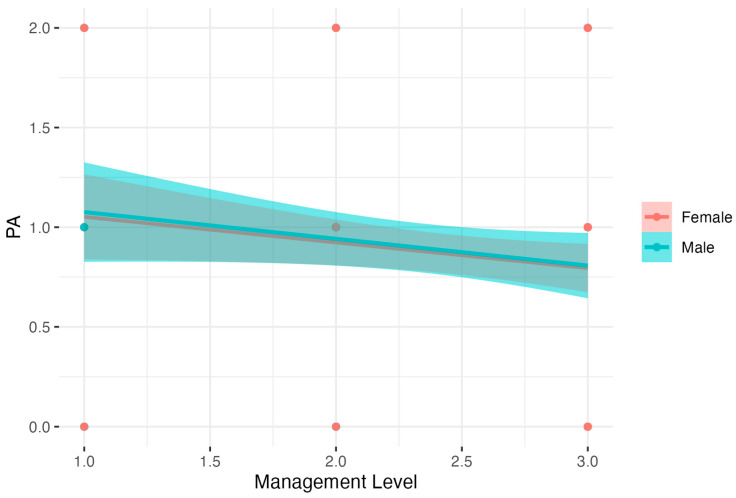
Gender as a moderator of the relationship between Management Level and Physical Activity.

**Figure 6 nutrients-15-04234-f006:**
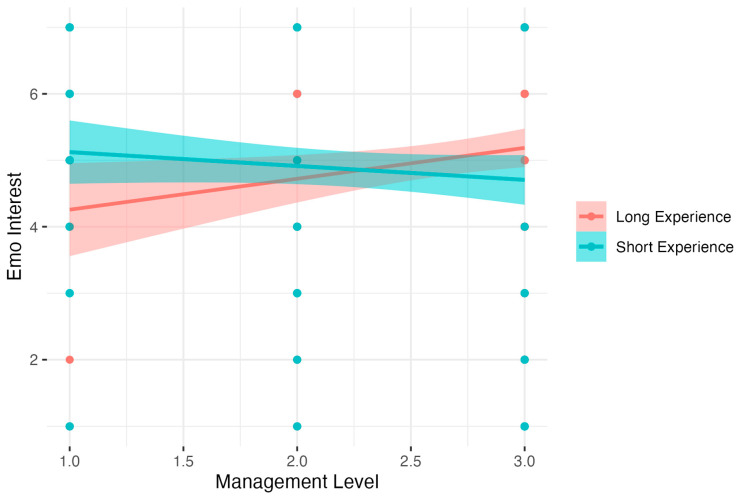
Experience as a moderator of the relationship between Management Level and Emotional Interest.

**Figure 7 nutrients-15-04234-f007:**
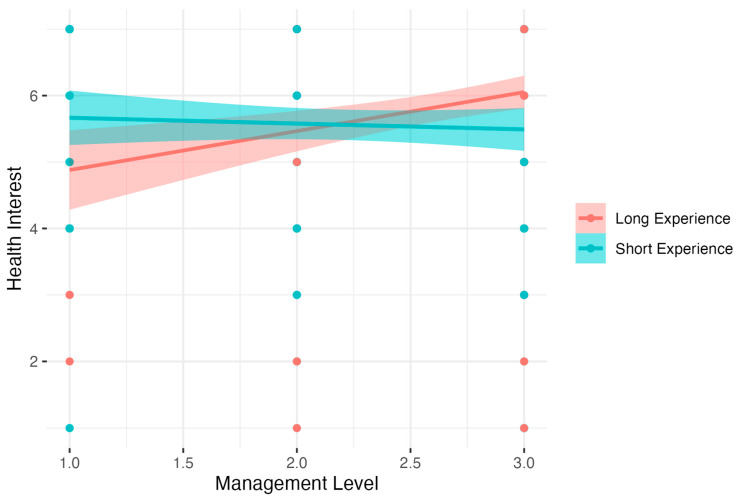
Experience as moderator of the relationship between Management Level and Health Interest.

**Figure 8 nutrients-15-04234-f008:**
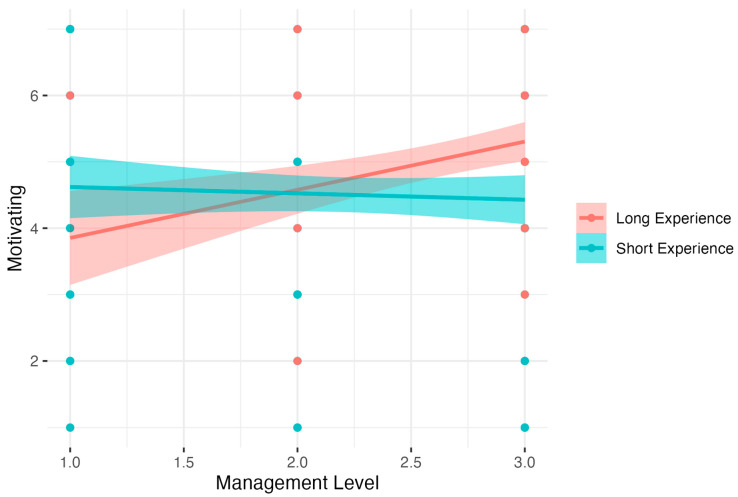
Experience as a moderator of the relationship between Management Level and Motivating.

**Figure 9 nutrients-15-04234-f009:**
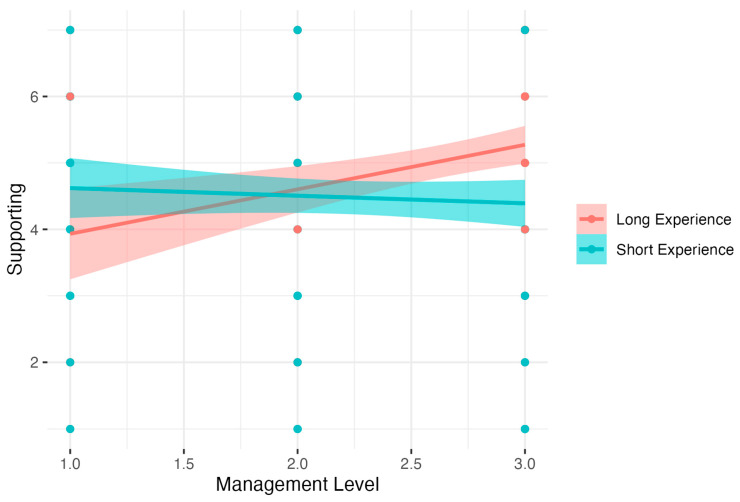
Experience as a moderator of the relationship between Management Level and Supporting.

**Figure 10 nutrients-15-04234-f010:**
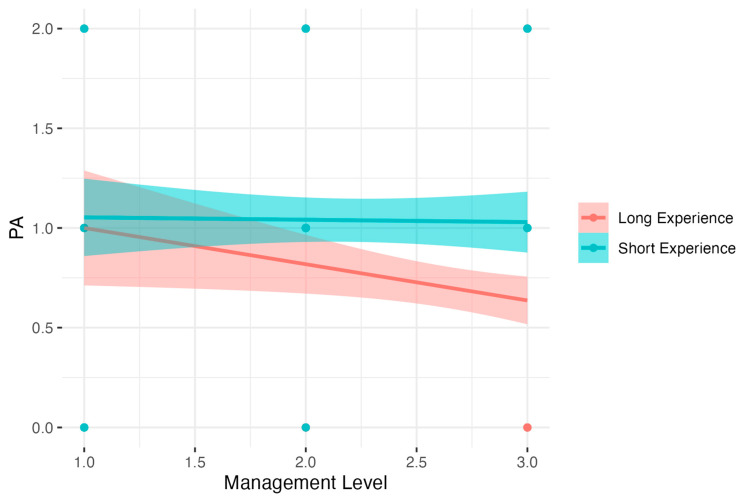
Experience as a moderator of the relationship between Management Level and Physical Activity.

**Table 1 nutrients-15-04234-t001:** Sample characteristics.

Characteristic	Female, N = 222 ^1^	Male, N = 126 ^1^	*p*-Value ^2^
Dieting (EAT-26)	4.46 (0.78)	4.64 (0.74)	0.040
Bulimia and Food Preoccupation (EAT-26)	5.34 (0.59)	5.34 (0.70)	0.359
Oral Control (EAT-26)	5.11 (0.55)	5.07 (0.65)	0.975
Physical Activity	0.86 (0.75)	0.90 (0.70)	0.641
Experience	23.82 (9.82)	24.03 (9.27)	0.996
Experience as a Manager	11.91 (8.28)	13.29 (9.29)	0.258
Number of Subordinates	45.49 (117.00)	77.72 (359.85)	0.768
Management Level			0.097
First line manager	36/222 (16%)	22/126 (17%)	
Mid_level manager	49/222 (22%)	40/126 (32%)	
Top manager	137/222 (62%)	64/126 (51%)	
Supporting (soft skill)	4.91 (1.69)	4.45 (1.70)	0.013
Emo Interest (soft skill)	5.21 (1.66)	4.47 (1.80)	<0.001
Motivating (Soft skill)	5.05 (1.72)	4.27 (1.76)	<0.001
Health Interest (soft skill)	5.87 (1.42)	5.39 (1.62)	0.005
Age	47.99 (9.54)	48.58 (9.88)	0.706
Education			<0.001
Higher_Bachelor	11/222 (5.0%)	8/126 (6.3%)	
Higher_Master’sdegree	190/222 (86%)	84/126 (67%)	
Medium	4/222 (1.8%)	4/126 (3.2%)	
PhDorhigher	17/222 (7.7%)	30/126 (24%)	
BMI	24.62 (4.87)	26.92 (3.95)	<0.001
BMI cat			<0.001
Normal Weight	134/222 (60%)	34/126 (27%)	
Obesity	32/222 (14%)	25/126 (20%)	
Overweight	50/222 (23%)	65/126 (52%)	
Underweight	6/222 (2.7%)	2/126 (1.6%)	

^1^ Mean (SD); n/N (%). ^2^ Wilcoxon rank sum test; Pearson’s Chi-squared test; Fisher’s exact test.

**Table 2 nutrients-15-04234-t002:** Estimates of regression models with interaction terms.

Dependent Variable	Variables in Model	*B*	s.e.	*t*	LCI	UCI	*p*	β	LCI2	UCI2
B and F Preocup (1)	Intercept	5.25	0.14	36.76	4.97	5.53	<0.001			
	GenderMale	−0.46	0.23	−2.00	−0.91	−0.01	<0.05	−0.35	−0.8	0.1
	Management Level	0.03	0.06	0.59	−0.08	0.14	>0.05	0.04	−0.07	0.15
	GenderMale ∗ Management Level	0.20	0.09	2.20	0.02	0.39	<0.05	0.39	0.21	0.57
Dieting (2)	Intercept	4.74	0.17	27.29	4.40	5.09	<0.001			
	GenderMale	−0.59	0.28	−2.09	−1.14	−0.03	<0.05	−0.37	−0.92	0.19
	Management Level	−0.12	0.07	−1.71	−0.25	0.02	<0.10	−0.11	−0.25	0.02
	GenderMale ∗ Management Level	0.32	0.11	2.88	0.10	0.54	<0.01	0.51	0.29	0.73
Oral Control (3)	Intercept	5.14	0.13	38.38	4.88	5.40	<0.001			
	GenderMale	−0.45	0.22	−2.06	−0.87	−0.02	<0.05	−0.36	−0.79	0.06
	Management Level	−0.02	0.05	−0.30	−0.12	0.09	>0.05	−0.02	−0.12	0.08
	GenderMale ∗ Management Level	0.18	0.09	2.04	0.01	0.35	<0.05	0.36	0.19	0.53
Supporting (4)	Intercept	4.83	0.38	12.61	4.08	5.58	<0.001			
	GenderMale	−1.77	0.62	−2.86	−2.99	−0.55	<0.01	−0.5	−1.72	0.72
	Management Level	0.04	0.15	0.25	−0.26	0.33	>0.05	0.02	−0.28	0.31
	GenderMale ∗ Management Level	0.55	0.25	2.24	0.07	1.04	<0.05	0.39	−0.09	0.88
Physical Activity (5)	Intercept	1.18	0.17	7.10	0.85	1.51	<0.001			
	GenderMale	0.01	0.27	0.04	−0.52	0.54	>0.05	0.01	−0.52	0.53
	Management Level	−0.13	0.06	−1.99	−0.26	0.00	<0.05	−0.13	−0.26	−0.01
	GenderMale ∗ Management Level	0.00	0.11	−0.01	−0.21	0.21	>0.05	0	−0.21	0.21
Emo interest (6)	Intercept	6.52	0.70	9.32	5.14	7.89	<0.001			
	Experience	−0.08	0.03	−2.60	−0.14	−0.02	<0.01	−0.45	−0.51	−0.39
	Management Level	−0.72	0.30	−2.38	−1.32	−0.13	<0.05	−0.32	−0.91	0.28
	Experience ∗ Management Level	0.04	0.01	2.80	0.01	0.06	<0.01	0.66	0.64	0.69
Health interesting (7)	Intercept	6.74	0.60	11.22	5.56	7.93	<0.001			
	Experience	−0.07	0.03	−2.62	−0.12	−0.02	<0.01	−0.45	−0.5	−0.39
	Management Level	−0.62	0.26	−2.36	−1.13	−0.10	<0.05	−0.31	−0.82	0.2
	Experience ∗ Management Level	0.04	0.01	3.26	0.01	0.06	<0.01	0.76	0.74	0.78
Motivating (8)	Intercept	5.85	0.70	8.36	4.47	7.23	<0.001			
	Experience	−0.08	0.03	−2.53	−0.14	−0.02	<0.05	−0.43	−0.49	−0.36
	Management Level	−0.73	0.30	−2.42	−1.33	−0.14	<0.05	−0.31	−0.91	0.28
	Experience ∗ Management Level	0.04	0.01	3.40	0.02	0.07	<0.001	0.79	0.77	0.81
Supporting (9)	Intercept	5.51	0.68	8.15	4.18	6.84	<0.001			
	Experience	−0.06	0.03	−1.96	−0.12	0.00	<0.10	−0.33	−0.39	−0.27
	Management Level	−0.60	0.29	−2.06	−1.18	−0.03	<0.05	−0.27	−0.85	0.31
	Experience ∗ Management Level	0.04	0.01	2.90	0.01	0.06	<0.01	0.68	0.65	0.7
PA (10)	Intercept	1.04	0.19	5.33	0.65	1.42	<0.001			
	Experience	0.01	0.02	0.77	−0.02	0.05	>0.05	0.15	0.12	0.18
	Management Level	−0.03	0.08	−0.35	−0.19	0.13	>0.05	−0.03	−0.19	0.13
	Experience ∗ Management Level	−0.01	0.01	−1.23	−0.02	0.00	>0.05	−0.27	−0.29	−0.26

## Data Availability

The data is available by contacting the second co-author of the study.

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
