# Peer review of "Gender and Work Experience as Moderators of Relations between Management Level, Physical Activity, Eating Attitudes, and Social Skills of Managers during the COVID-19 Pandemic"

_nutrients, 2023, doi:10.3390/nu15194234_

Round 1

Reviewer 1 Report

Summary of Findings:
The study found that male managers exhibited lower scores on eating attitude scales compared to females, but males in high managerial positions showed higher levels of concerns related to body weight, body shape, and eating. Both genders had similar levels of physical activity. Experience and management level correlated with certain soft skills, with more experienced and higher-ranking managers demonstrating stronger emotional and health-related concerns, as well as greater supportiveness and motivation towards their employees. However, these factors did not significantly impact physical activity levels.

Assessment:

The paper investigates the influence of gender, experience, and management level on managers' eating attitudes, physical activity, and soft skills during the COVID-19 pandemic. The introduction provides a comprehensive background, references, and clear research questions. The cited references are relevant to the research, and the research design appears appropriate for the study's objectives. The methods are adequately described, and the results are clearly presented. However, the conclusions could be more explicitly tied to the results. The English literacy is of good quality.

Author Response

Dear Editor and Reviewers,

Thank you for your review and valuable opinion and the opportunity to resubmit our paper. We modified the paper according to your suggestions. We hope you will find it a high-quality scientific manuscript compatible with the high standards of the Nutrients MDPI.

.

Responses to Reviewer 1 Comments

Reviewer 1

Summary of Findings:

The study found that male managers exhibited lower scores on eating attitude scales compared to females, but males in high managerial positions showed higher levels of concerns related to body weight, body shape, and eating. Both genders had similar levels of physical activity. Experience and management level correlated with certain soft skills, with more experienced and higher-ranking managers demonstrating stronger emotional and health-related concerns, as well as greater supportiveness and motivation towards their employees. However, these factors did not significantly impact physical activity levels.

Assessment:

The paper investigates the influence of gender, experience, and management level on managers' eating attitudes, physical activity, and soft skills during the COVID-19 pandemic. The introduction provides a comprehensive background, references, and clear research questions. The cited references are relevant to the research, and the research design appears appropriate for the study's objectives. The methods are adequately described, and the results are clearly presented. However, the conclusions could be more explicitly tied to the results. The English literacy is of good quality.

Response: We have introduced a new subsection, "2.3. Statistical Analysis," to enhance the paper's clarity. Additionally, we have included Table 1, which presents sample statistics, including calculated p-values, for both males and females. This table also encompasses EAT-26 measurements, BMI (along with categorization based on norms), and age. Lines 231-265. As you recommended, we supported the Discussion with the Conclusion (lines 535-550).

Reviewer 2 Report

The presented work is too long, it should be considered whether the answer to the research question “Are experience and management level mediators of managers' PA levels and soft skills? “can be presented in another publication?

The introduction is too extensive, especially in the description of theories of leadership.

Materials and Methods section

Have gender differences in the age of the respondents been checked?

There are some inaccuracies in the description regarding the meeting of the survey group (Line 132 Data were collected in 2021 and 2022 but in Line 168 The online survey took place from January 2022 to December 2022)

In the material and methods section, the statistical analysis subchapter should be included, which will improve the description and presentation of the results.

In the result section of Table 1, the results should be presented by gender

Information regarding place of residence and assessment of nutritional status (e.g. interpretation of the respondents' BMI) is not presented in Table 1?

Results do not provide information on the final result of the EAT-26 test taking into account gender.

Discussion at work is too general and needs improvement.

Author Response

Dear Editor and Reviewers,

Thank you for your review and valuable opinion and the opportunity to resubmit our paper. We have studied carefully all comments to prepare the revision of our paper. We hope you will find it a high-quality scientific manuscript compatible with the high standards of the Nutrients MDPI.

Responses to Reviewer 2 Comments

Materials and Methods section

Have gender differences in the age of the respondents been checked?

There are some inaccuracies in the description regarding the meeting of the survey group (Line 132 Data were collected in 2021 and 2022 but in Line 168 The online survey took place from January 2022 to December 2022)

Point 1: The presented work is too long, it should be considered whether the answer to the research question “Are experience and management level mediators of managers' PA levels and soft skills? “can be presented in another publication?

Response 1: Our study aimed to comprehensively investigate the impact of gender, experience, and management level on managers' behaviors and soft skills during the pandemic. Formulating specific research questions allowed us to achieve clarity in addressing this multifaceted scope.

By addressing all research questions in one publication, we maintain consistency with our comprehensive research objectives, ensuring a holistic examination of the topic. This approach enhances the practical relevance of our findings for organizations and professionals, and it contributes significantly to the existing body of knowledge.

In conclusion, our decision to present all findings in a single publication aligns with our research objectives, enhances practical relevance, and contributes to the scientific discussion in the field. We appreciate the reviewer's feedback and believe this approach best serves the study's goals.

Point 2: The introduction is too extensive, especially in the description of theories of leadership.

Response 2: We shortened the description of theories of leadership to one sentence and we moved it to the last part of the third paragraph. Lines 118-122

Materials and Methods section

Thank you for your valuable comments. We agree with your assessment of the text structure and certain additional statistical considerations, and as a result, we have made some corrections.

Point 3: There are some inaccuracies in the description regarding the meeting of the survey group (Line 132 Data were collected in 2021 and 2022 but in Line 168 The online survey took place from January 2022 to December 2022)

Response 3: Thank you very much for your accuracy. We corrected this mistake. Line 165.

Point 4: In the material and methods section, the statistical analysis subchapter should be included, which will improve the description and presentation of the results.

In the result section of Table 1, the results should be presented by gender

Have gender differences in the age of the respondents been checked?

Information regarding place of residence and assessment of nutritional status (e.g. interpretation of the respondents' BMI) is not presented in Table 1? Results do not provide information on the final result of the EAT-26 test taking into account gender.

Response 4: We have introduced a new subsection, "2.3. Statistical Analysis," to enhance the paper's clarity. Additionally, we have included Table 1, which presents sample statistics, including calculated p-values, for both males and females. This table also encompasses EAT-26 measurements, BMI (along with categorization based on norms), and age. Lines 231-265.

Regrettably, our study did not collect information regarding nutritional aspects of functioning or place of residence, and we also lack proxies for these variables. It's important to note that these variables were not within the scope of our original design assumptions.

Given that our data was sourced from respondents in various major cities across Poland, most of whom were employed by large companies, the variable of place of residence was intentionally excluded from our considerations, as there was minimal variation among our respondents in this regard. We supported the Discussion (limitations) with the information in lines 524-526, which sounds: “There are no data concerning the character of the corporations the respondents were working for as well as the place of residence, which could have provided meaningful context and thereby revealed other significant correlations”.

Point 5: Discussion at work is too general and needs improvement.

Response 5: We supported the Discussion with the Conclusion (lines 535-550) and two new references line 471.
